# Cell-based glycan arrays for probing glycan–glycan binding protein interactions

Jennie Grace Briard[1], Hao Jiang[2], Kelley W. Moremen[3], Matthew Scott Macauley[1,4] & Peng Wu [1]

Glycan microarrays provide a high-throughput means of profiling the interactions of glycan-binding proteins with their ligands. However, the construction of current glycan microarray platforms is time consuming and expensive. Here, we report a fast and cost-effective method for the assembly of cell-based glycan arrays to probe glycan–glycan-binding protein interactions directly on the cell surface. Chinese hamster ovary cell mutants with a narrow and relatively homogeneous repertoire of glycoforms serve as the foundation platforms to develop these arrays. Using recombinant glycosyltransferases, sialic acid, fucose, and analogs thereof are installed on cell-surface glycans to form cell-based arrays displaying diverse glycan epitopes that can be probed with glycan-binding proteins by flow cytometry. Using this platform, high-affinity glycan ligands are discovered for Siglec-15—a sialic acid-binding lectin involved in osteoclast differentiation. Incubating human osteoprogenitor cells with cells displaying a high-affinity Siglec-15 ligand impairs osteoclast differentiation, demonstrating the utility of this cell-based glycan array technology.

[1] Department of Molecular Medicine, The Scripps Research Institute, La Jolla, CA 92037, USA. [2] Key Laboratory of Marine Drugs, Ministry of Education and Qingdao National Laboratory for Marine Science & Technology and Shandong Provincial Key Laboratory of Glycoscience & Glycoengineering, School of Medicine and Pharmacy, Ocean University of China, 266003 Qingdao, China. [3] Complex Carbohydrate Research Center and the Department of Biochemistry and Molecular Biology, University of Georgia, Athens, GA 30602, USA. [4] Present address: Department of Chemistry, University of Alberta, Edmonton, AB T6G 2G2, Canada. Correspondence and requests for materials should be addressed to M.S.M. (email: macauley@ualberta.ca) or to P.W. (email: pengwu@scripps.edu)

Glycans decorate the cell surface of both eukaryotes and prokaryotes, and in mammalian cells are involved in a variety of physiological processes, including angiogenesis, fertilization, stem cell development, and neuronal development[1–3]. Changes in glycosylation patterns have also been shown to mark the onset of cancer and inflammation[2,3]. In many cases, glycans execute these cellular functions by interacting with glycan-binding proteins (GBPs). Therefore, there is enormous interest in understanding the structural basis of these interactions for the dissection of the mechanisms of glycan-mediated biological processes and for the development of new therapeutic agents to treat glycan-regulated disease. Unfortunately, it is challenging to probe glycan−GBP interactions in vivo because glycosylation is a post-translational modification not under direct genetic control. The dynamic process of glycosylation orchestrated by glycosylation enzymes results in heterogeneous glycoconjugates found on the cell surface and on secreted proteins[3].

Glycan microarrays were developed in response to the critical need for high-throughput methods to identify GBP interactions[4,5]. As highlighted in Transforming Glycoscience (section 5.1.1), these microarrays have been extensively employed to interrogate binding specificities of a diverse range of GBPs, determine dissociation constants, dissect binding energies, and assess multivalent and hetero-ligand binding[6]. Currently, most glycan arrays are constructed by coupling a chemically defined glycan to a solid support, such as a glass slide[4,5]. Such homogeneous glycans and derivatives are either synthesized[4] or purified from natural sources by multi-dimensional chromatography[7]. Several noteworthy drawbacks are associated with the current platforms. First, obtaining samples of pure, well-characterized oligosaccharides for the assembly of glycan arrays by chemical or chromatography-based purification is time consuming and can only be performed by a specialist. As such, glycosyltransferases are often employed in combination with chemical synthesis to facilitate the production of complex oligosaccharides[8]. However, only limited numbers of glycosyltransferases are present in carbohydrate chemists' toolbox. Therefore, many glycosidic linkages cannot be assembled in a straightforward manner. The second drawback is that the current glycan microarrays do not fully recapitulate the natural cell-surface environment on which glycans are presented. Indeed, Wong and co-workers have shown that the poor sensitivity of the conventional microarrays arises from their surface-generated pseudo-multivalent display[9]. To better mimic the natural multivalent presentation, several groups have developed creative strategies by attaching synthetic glycans to protein[10] or polymer scaffolds[11]. These approaches, however, also rely on the lengthy synthesis of complex glycans.

Here, we describe a method to chemoenzymatically install monosaccharides and their analogs directly on the cell surface to create in-solution, cell-based arrays displaying chemically defined peripheral glycan epitopes. The lectin-resistant Chinese hamster ovary (CHO) cell mutant Lec2 that expresses a narrow and relatively homogenous repertoire of glycoforms is employed as the foundation platform. With the conserved core glycan structures already expressed on the cell surface, the lengthy synthesis required to build complex carbohydrates is avoided. Using a handful of glycosyltransferases compatible with cell-surface glycosylation, sialic acid, fucose, and their analogs are introduced to these cells' peripheral glycans linkage specifically to form cell-based arrays displaying diverse glycan epitopes. We demonstrate the utility of these cell-based arrays to interrogate GBP specificities and ligand tolerance directly on the cell surface. This method is applied to high throughput screening for the identification of selective and high-affinity ligands of Siglecs, a family of sialic acid-binding immunoglobulin-type lectins that are differentially expressed primarily on immune cells. Using this approach, a high-affinity glycan ligand for Siglec-15 is discovered that can be used to modulate the differentiation of osteoclasts.

## Results

**Design and validation of cell-based glycan array strategy**. As proof-of-principle, we used the CHO glycosylation mutant Lec2 cells[12] to construct in-solution, cell-based glycan arrays displaying defined periphery glycans (Fig. 1a). Lec2 cells have an inactive CMP-sialic acid Golgi transporter. As a consequence, no sialylation occur without the donor substrate CMP sialic acid available in the Golgi. In addition, there are no active α1-2, α1-3, and α1-4 fucosyltransferases (FTs) and, therefore, their cell-surface N-glycans terminate with N-acetyllactosamine (LacNAc, Galβ1-4GlcNAc) (Fig. 2c)[12]. Lacking terminal fucose or sialic acid (Fig. 1a) makes Lec2 cells a rich source of acceptor substrates for FTs and sialyltransferases (STs). Accordingly, *Helicobacter pylori* α1-3FT[13–15], human α2-3ST (ST3Gal4)[16], and rat α2-6ST (ST6Gal1)[16–18] were employed to install fucose or sialic acid onto the cell surface in a linkage-specific manner. The addition of α1-3-linked fucose leads to the formation of Lewis X (Galβ1-4 (Fucα1-3)GlcNAc, Le[X]) (Fig. 1b). The addition of α2-6- or α2-3-linked sialic acid leads to the formation of Siaα2-6Galβ1-4GlcNAc or Siaα2-3Galβ1-4GlcNAc, respectively (Fig. 1d).

The cells displaying Le[X] were constructed by treating Lec2 cells with GDP-fucose (GDP-Fuc) (**1**) (500 μM) and α1-3FT in Hanks buffered salt solution (HBSS) for 10 min at 37 °C (Fig. 2a). Lec8 CHO cells that do not express LacNAc due to a mutation in the UDP-Gal Golgi transporter were used as a negative control[19]. The modified cells were then probed with fucoside binding proteins or antibodies, including anti-stage-specific embryonic antigen-1 (anti-SSEA-1 or anti-Le[X]), *Aleuria aurantia* lectin (AAL), and *Lotus tetragonolobus* lectin (Lotus A) as FITC conjugates[20]. Flow cytometry reveals that anti-SSEA-1, AAL, and Lotus A all exhibited strong binding to cells displaying the Le[X] epitope (Fig. 2b). In contrast, only background binding was observed with non-fucosylated Lec2 cells and control Lec8 cells.

The cells displaying Siaα2-3Galβ1-4GlcNAc and Siaα2-6Galβ1-4GlcNAc were constructed by treatment of Lec2 cells with CMP-Neu5Ac (**6**) (Fig. 2a) and ST3Gal4 or ST6Gal1, respectively, for 1 h at 37 °C. Again, Lec8 CHO cells were used as a negative control. These two cells were first probed with FITC-conjugated *Erythrina cristagalli* lectin (ECL) that is specific for terminal galactose and N-acetylgalactosamine and as such, bound strongly to unmodified Lec2 CHO cells (Fig. 2c)[21]. Consistent with the specificity of ECL, the cells with installed sialic acid resulted in reduced binding (Fig. 2c). The cells were then probed with FITC-conjugated sialoside binding proteins, *Sambucus nigra* (SNA-I) and *Maackia amurensis* lectin (MAL-II), specific for α2-6- and α2-3-linked sialic acids, respectively[20]. SNA-I strongly bound to cells displaying α2-6-linked sialic acid, whereas no binding was observed for cells with α2-3-linked sialic acid (Fig. 2c). For MAL-II, the opposite was observed (Fig. 2c). In these experiments, as little as 1 μg/mL of GBPs was used. Together, these results provide strong evidence that our approach for installing monosaccharides to construct ligands for probing their interactions with GBPs on cell surfaces is specific and sensitive.

**Construct glycan arrays displaying unnatural epitopes**. Screening glycan libraries with unnatural substituents serves as a powerful means to probe the binding pocket of a GBP especially when there is limited structural information available. To apply cell-based glycan arrays to profile the substrate tolerance of GBPs, we constructed three arrays displaying Le[X], Siaα2-3Galβ1-4GlcNAc, Siaα2-6Galβ1-4GlcNAc and their structurally related

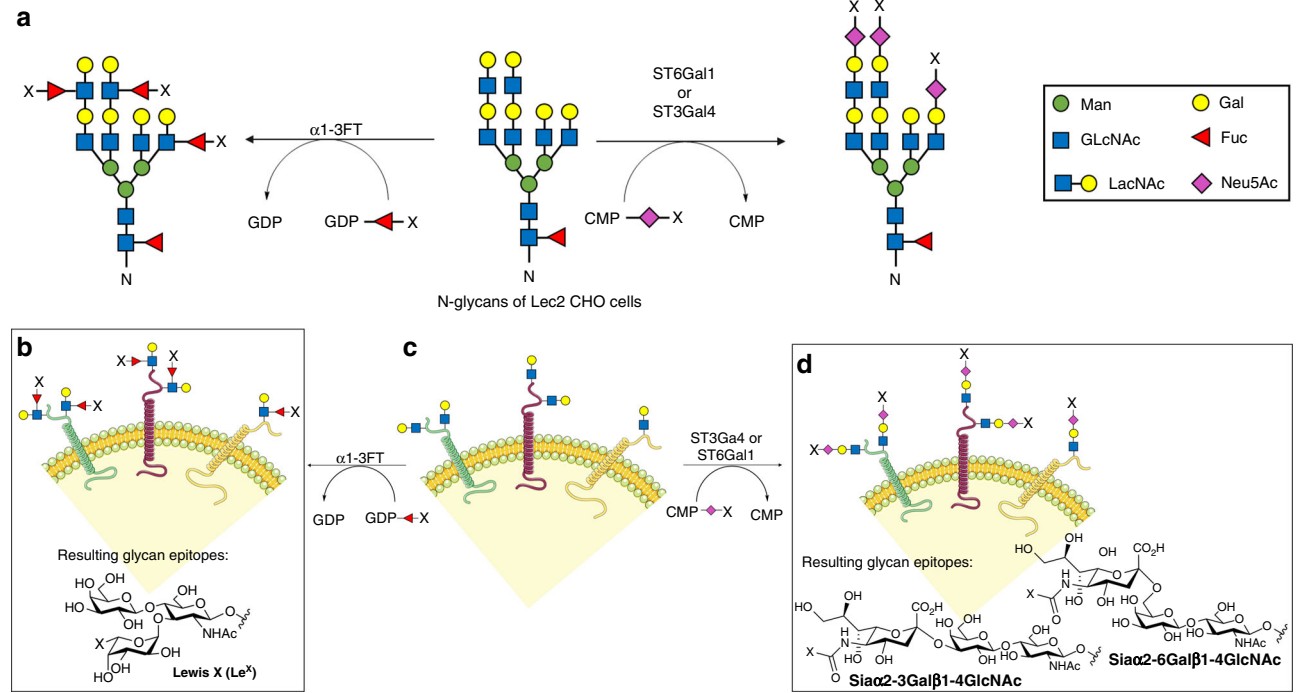

**Fig. 1** The workflow for constructing cell-based glycan arrays. **a** Distinct glycan epitopes are installed onto N-glycans of Lec2 CHO cells via sialyltransferase-mediated cell-surface in situ sialylation using the CMP-Neu5Ac donor (or its analogs) or fucosyltransferase-mediated cell-surface in situ fucosylation using the GDP-Fuc donor (or its analogs). **b** Cell-based glycan arrays displaying Le$^X$ or its derivatives. **c** N-glycans of Lec2 CHO cells. **d** Cell-based glycan arrays displaying α2-3- or α2-6-linked sialosides, or their derivatives. X represents unnatural substituents introduced at fucose C-5 or NeuNAc acetyl position

derivatives using Lec2 CHO cells as the starting platform. The array displaying Le$^X$ and its derivatives was constructed using 500 μM GDP-Fuc (**1**) or GDP-Fuc analogs (**2**−**5**) (Fig. 3a). Our previous studies revealed that while α1-3FT is highly specific for acceptor glycans possessing LacNAc, it has relaxed donor substrate specificity and is able to accept GDP-Fuc donors with a wide variety of functional groups at the C5 position of fucose[15]. Kinetic analysis of the enzymatic transformation showed only minor differences (2.8, 8.4, 6.2, 24 μM for **2**–**5** vs. 16 μM for GDP-Fuc (**1**)) in $K_m$ values for GDP-Fuc donors with functional groups at C5. Furthermore, there was no appreciable influence on the $V_{max}$ (Supplementary Fig. 1). Therefore, we chose 500 μM as the concentration for all donor substrates for surface fucosylation, which was well above the saturated substrate concentration.

The arrays displaying Siaα2-3Galβ1-4GlcNAc, Siaα2-6Galβ1-4GlcNAc and their derivatives were constructed using CMP-Neu5Ac (**6**) or CMP-Neu5Ac analogs (**7**–**9**) (Fig. 3b). Upon sialylation by ST3Gal4 and ST6Gal1, Lec2 CHO cells showed a reduction in ECL binding indicating donor transfer regardless of the sialic analog used (Supplementary Fig. 2). This observation is consistent with an earlier discovery showing that ST6Gal1 was promiscuous for C5-modified CMP-Neu5Ac analogs and could even tolerate large C5 moieties like biotin[22,23].

The array displaying Le$^X$ and its derivatives was probed with fucoside binding proteins, SSEA-1, AAL, and Lotus A. The arrays displaying Siaα2-3Galβ1-4GlcNAc, Siaα2-6Galβ1-4GlcNAc and their derivatives were probed with SNA-I and MAL-II. Interestingly, we discovered that this panel of GBPs exhibited distinct binding tolerances to unnatural glycan derivatives. Anti-SSEA-1, also known as anti-Le$^X$ antibody, was able to tolerate an alkyne (**2**) and a hydroxyl group (**4**) at the C5 position of fucose, but not an azide (**3**) or methoxy (**5**) functional group, which abrogated interaction (Fig. 3c). All fucose analogs assessed resulted in a

reduction in AAL binding but did not entirely abolish this interaction. Lotus A was very sensitive to C5 modification and the interaction was completely blocked by any modification to the C5 position of fucose. On the other hand, SNA-I and MAL-II were not sensitive to sialic acid modifications at C5, but demonstrated linkage-specificity (Fig. 3d). SNA-I bound to all sialic acid analogs when attached in an α2-6-linkage. Alternatively, MAL-II tolerated all analogs attached in an α2-3-linkage.

**Screen high-affinity Siglec-15 ligands.** Having demonstrated that our cell-based glycan array was capable of detecting GBP binding specificity and substrate tolerance, we next assessed the applicability of this method to identify high-affinity Siglec ligands. Siglecs comprise a family of 15 members of sialic acid-binding receptors that are differentially expressed on immune cells[24–26]. Because of the restricted expression of Siglecs to one or a few immune cell types, Siglecs are attractive targets for cell-directed therapies in immune-cell-mediated diseases[26–29]. Furthermore, Siglecs are endocytic receptors allowing efficient uptake of therapeutic agents conjugated to an antibody or glycan ligand[30–33]. High-affinity and highly selective glycan ligands represent attractive alternatives to antibody-based therapeutics with several notable advantages. Notably, nanoparticles bearing sialoside ligands for delivery of therapeutic cargo have shown promise for targeting Siglecs in vivo[34]. To date, high-affinity and selective glycan ligands have been discovered for approximately half of human Siglecs using conventional glycan microarray technology[35–38].

We sought to determine the applicability of our cell-based array technique to discover high-affinity and selective glycan ligands for Siglecs. One Siglec of interest where high-affinity and selective ligands have not been discovered is Siglec-15. Siglec-15

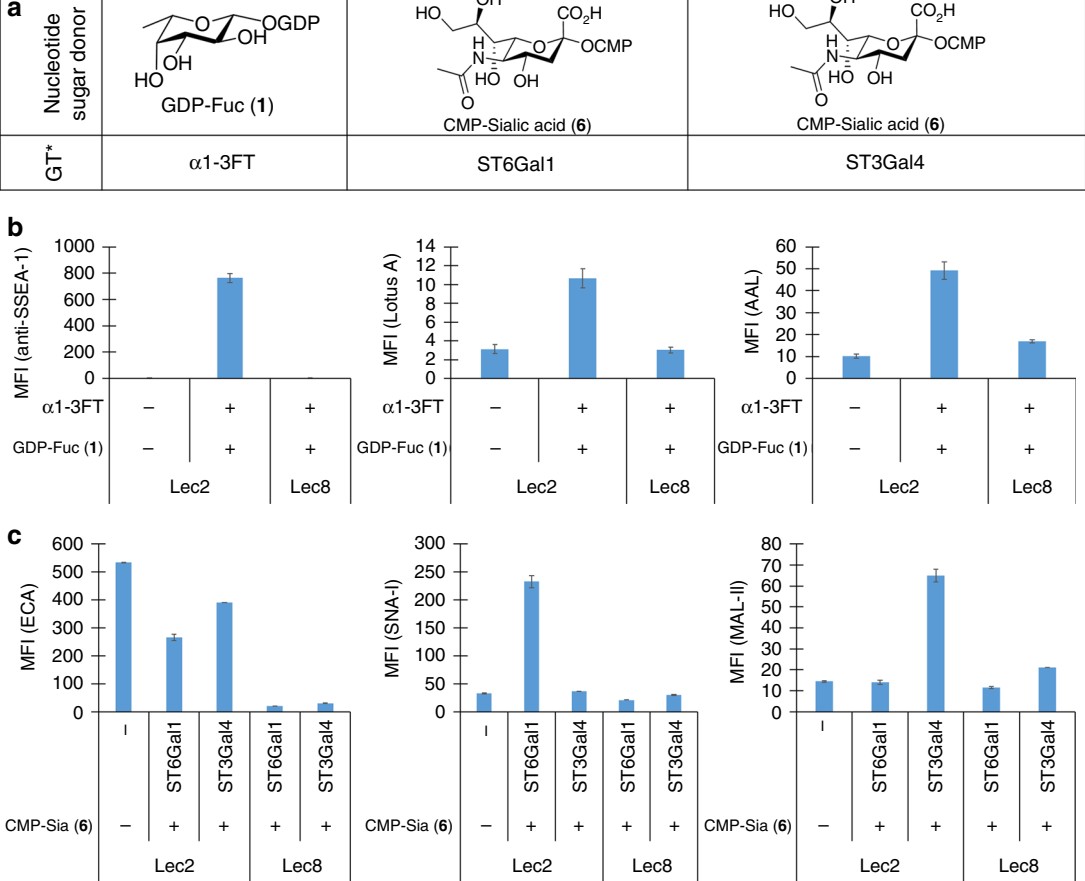

**Fig. 2** Validate the specificity of in situ glycosylation strategy. **a** Nucleotide sugar donors and glycosyltransferases used to form three glycan epitopes (Le$^X$, Siaα2-3Galβ1-4GlcNAc, Siaα2-6Galβ1-4GlcNAc) on the cell surface (*GT = glycosyltransferase). **b** Flow cytometry data presented in bar graph for validating cell-surface in situ fucosylation. CHO cells were modified via in situ fucosylation using the donor GDP-Fuc (**1**) and α1-3FT . The modified cells were probed with a fluorescently labeled lectin or antibody and analyzed by flow cytometry ($n = 3$, error bars are reported as the standard error of the mean (SEM)). **c** Flow cytometry data presented in bar graph for validating in situ sialylation. CHO cells were modified via in situ sialylation using the donor CMP-Sia (**6**) and the sialyltransferase ST6Gal1 or ST3Gal4. The modified cells were probed with a fluorescently labeled lectin and analyzed by flow cytometry ($n = 3$, error bars are reported as SEM)

becomes expressed within days of RANKL (receptor activator of nuclear factor κβ ligand)-induced differentiation of osteoclast precursors into osteoclasts[39–43]. Studies have demonstrated that Siglec-15 positively regulates osteoclast differentiation and, consistent with this finding, loss of Siglec-15 results in impaired osteoclast differentiation and osteopetrosis in Siglec-15-deficient mice[40]. Therefore, Siglec-15 plays a critical role in bone remodeling and is a considerable potential therapeutic target for osteoporosis.

From what is known, Siglec-15 prefers α2-6-linked sialic acid residues in native ligands[43]. As several high-affinity Siglec ligands have been developed with functional modifications at the C5 position of Neu5Ac, we focused on this position on sialic acid to create chemical diversity[35–38]. To do so, we employed ST6Gal1 to install an α2-6-linked sialic acid bearing a bioorthogonal alkyne tag at C5 onto the cell surface of Lec2 cells (Fig. 4a). The alkyne tagged cells were then derivatized by reaction with a 40-member azide library (Supplementary Table 1) via biocompatible Cu(I)-catalyzed azide-alkyne cycloaddition (CuAAC) (Fig. 4a)[44] to form a cell-based glycan array . The library was chosen to contain structures with a variety of different functional groups to cover a range of chemical space. The resulting cell-based glycan array was probed with a recombinant Siglec-15-Fc chimera pre-complexed to APC-conjugated anti-human IgG1. From this screen, several

high-affinity glycan ligands for Siglec-15 were discovered (Fig. 4b). Three structures (derived from azides A13, A31, and A37) resulted in an increase in Siglec-15-Fc binding greater than 15-fold compared to Lec2 CHO cells sialylated with native sialic acid (**6**).

To determine if sialic acid derivatives functionalized with A13 and A31 still serve as high-affinity ligands for Siglec-15 when displayed on the cell surface in different linkages, CMP-SiaA13 and CMP-SiaA31 (**10** and **11**; Fig. 4c) were synthesized and used as the donor substrates for cell-surface in situ sialylation mediated by three sialyltransfereases, ST6Gal1, ST3Gal4, and ST3Gal1, followed by Siglec-15-Fc binding. First, the ability of each enzyme to tolerate the alkyne-tagged CMP-SiaPoc (**7**) and CMP-Neu5Ac analogs with larger functional groups (e.g. biotin) at C5 was verified by the two-step (transfer CMP-SiaPoc in step 1 and react with biotin-azide in step 2) and one-step (directly transfer a C5 biotin functionalized Neu5Ac analog) labeling, respectively, and followed by detection with streptavidin-APC (Supplementary Fig. 3). The one-step procedure was found to be more efficient than the two-step method for all sialyltransferases, consistent with previous reports[23]. Lec2 CHO cells were then subjected to the one-step labeling with **10** and **11** by ST6Gal1, ST3Gal4, or ST3Gal1. ST6Gal1 and ST3Gal4 install CMP-Neu5Ac analogs on terminal LacNAc residues of N-glycans resulting in a display of

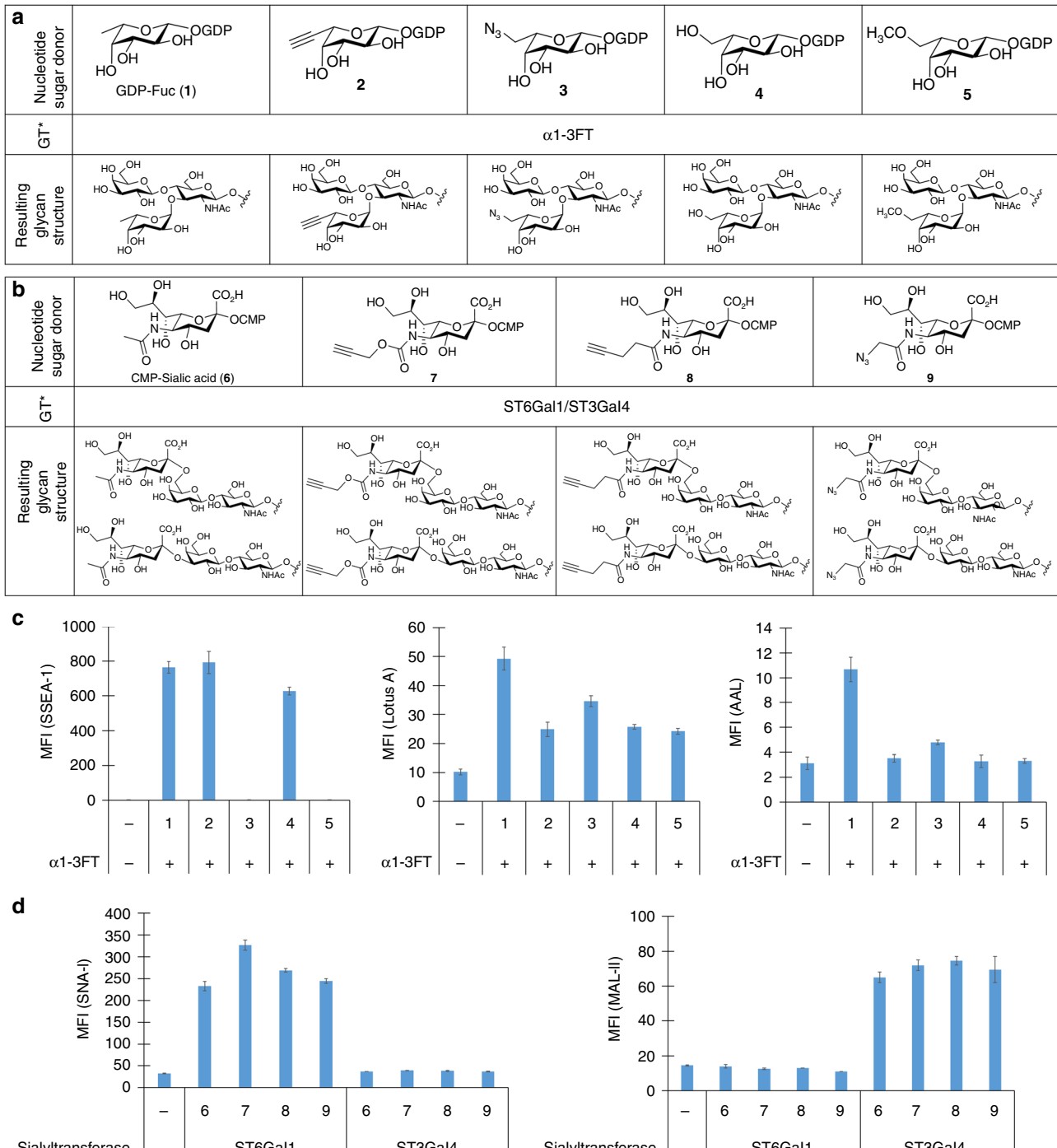

**Fig. 3** Use cell-based glycan arrays to profile GBP substrate scope. **a** GDP-Fuc(**1**), GDP-Fuc analogs (**2–5**) and resulting glycan structures after in situ cell-surface fucosylation. **b** CMP-sialic acid (**6**), CMP-sialic acid analogs (**7–9**) and resulting glycan structures after cell-surface sialylation. **c** Flow cytometry data presented in bar graph for profiling substrate tolerance of fucose-binding lectins and antibodies. CHO cells were modified via in situ fucosylation using α1-3FT and the donor substrate GDP-Fuc or its analogs. The modified cells were probed with a fluorescently labeled lectin or antibody and analyzed by flow cytometry ($n = 3$, error bars are reported as SEM). **d** Flow cytometry data presented in bar graph for profiling substrate tolerance of sialic acid-binding lectins. CHO cells were modified via in situ sialylation using the sialyltransferase ST6Gal1 or ST3Gal4 in the presence of the donor substrate CMP-Sia or its analogs. The modified cells were probed with a fluorescently labeled lectin and analyzed by flow cytometry ($n = 3$, error bars are reported as SEM)

α2-6- or α2-3-**10** or -**11** sialosides, respectively, whereas ST3Gal1 is responsible for transferring Neu5Ac analogs to the galactose of Galβ(**1**-**3**)GalNAc on O-glycans to form α2-3-linked sialosides (Fig. 4c)[13,37,45]. Siglec-15 bound to both α2-6-**10** and α2-3-**10** on N-glycans with similar affinity (Fig. 4d). By contrast, Siglec-15

preferred sialoside α2-6-**11** on N-glycans. Interestingly, Siglec-15 did not bind to either **10** or **11** derived sialosides when displayed in α2-3-linkages on O-glycans.

Many high-affinity glycan ligands for Siglecs lack the necessary specificity required for downstream applications[35]. To examine

the specificity of the Siglec-15 glycan ligands discovered here, we analyzed the binding of a panel of human Siglec-Fc chimeras to Lec2 CHO cells displaying **α2-6-10** or **α2-6-11** (Fig. 4e). Both **α2-6-10** and **α2-6-11** were found to be high-affinity ligands for Siglec-2, with a 123- and 81-fold increase in Siglec-2-Fc binding

compared to the negative control, respectively. Cells displaying **α2-6-10** also showed weak binding to Siglec-10-Fc. Subsequently, we screened the same panel of Siglecs using cells displaying **α2-3-10** and **α2-3-11** (Fig. 4f). Interestingly, both **α2-3-10** and **α2-3-11** showed dramatically increased specificity for Siglec-15—the

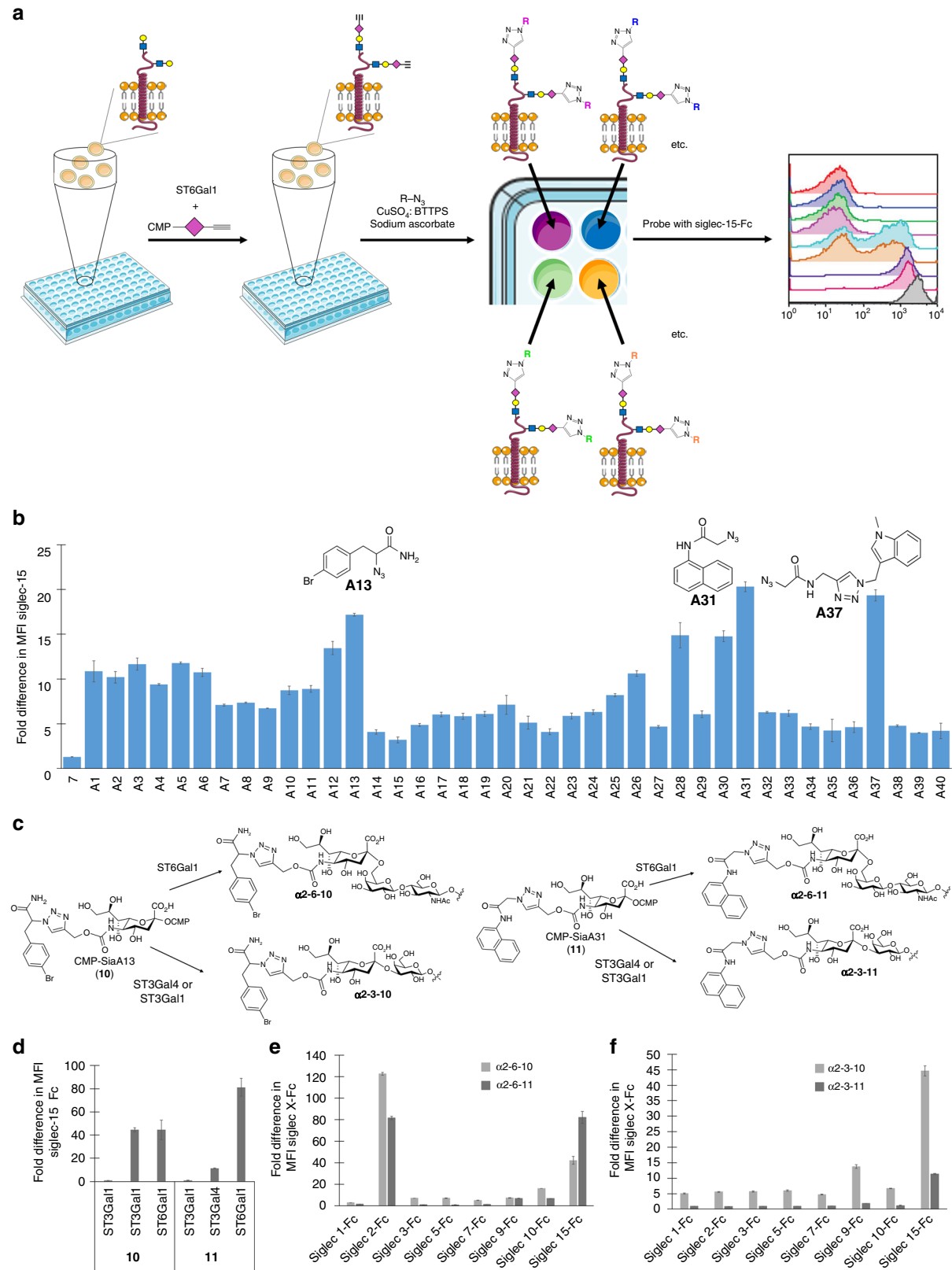

interaction with Siglec-2 was completely abolished. Although cells displaying **α2-3-10** still exhibited weak binding to Siglec-9, cells displaying **α2-3-11** showed remarkable specificity for Siglec-15 with negligible binding to all other Siglec-Fc chimeras assessed.

**High-affinity Siglec-15 ligands engage in cell−cell interactions.** With the identification that **10** could serve as the donor substrate to form high-affinity Siglec-15 ligands on the Lec2 cell surface, we then evaluated if these ligands could enforce interaction of modified cells with Siglec-15-expressing cells. Toward this end, we stained **α2-3-10** and **α2-6-10** displaying Lec2 CHO cells with CellTracker orange (CTO) as their identity marker. Unlabeled Lec2 CHO cells and cells labeled with the native ligand, **α2-6-6**, were used as controls. Subsequently, the Lec2 CHO cells stained with CTO were mixed with Siglec-15-expressing CHO cells that express extracellular Siglec-15 fused to intracellular GFP in a 100:1 ratio. The formation of cell clusters was determined by double staining and analyzed by flow cytometry and fluorescence microscopy (Fig. 5). Compared to the control non-sialylated Lec2 cells and cells modified with natural Neu5Ac, Lec2 cells modified with **10** induced significantly more cluster formation with Siglec-15-expressing cells.

While **α2-6-10** was found to also bind to Siglec-2 weakly, osteoclasts and human osteoprogenitor cells (HOPs; CD14$^+$ peripheral blood mononuclear cells) do not express Siglec-2. Therefore, after confirming that cells displaying **α2-6-10** could engage in cell−cell interactions via Siglec-15 binding, we assessed if Lec2 CHO cells displaying **α2-6-10** also could block osteoclast formation of RANKL -treated HOPs. RANKL, also known as osteoclast differentiation factor, is a cytokine secreted by osteoblasts that serves to activate osteoclasts, which are critically involved in bone resorption[42]. To determine if Lec2 CHO cells displaying **α2-6-10** could block osteoclast formation from HOPs, HOPs were isolated and plated. Osteoclast differentiation was induced by the addition of human macrophage colony-stimulating factor (hM-CSF) and RANKL[46]. Cells were allowed to differentiate for 7 days. On day 5, Lec2 CHO cells labeled with the high-affinity Siglec-15 glycan ligand, **α2-6-10**, or the native ligand, **α2-6-6**, were added to the plated cells. In these experiments, unlabeled Lec2 CHO cells were used as the negative control. On day 7, the formation of osteoclasts, which express abundant tartrate-resistant acid phosphatase (TRAP), was measured using the TRAP staining assay[47]. Osteoclasts were identified as TRAP-positive with 3 or more nuclei[46,47]. Remarkably, the addition of Lec2 CHO cells labeled with the high-affinity Siglec-15 ligand on day 5 resulted in over 50% reduction in osteoclast formation compared to the addition of unlabeled Lec2 CHO cells and HOPs treated with RANKEL and and hM-CSF only (Fig. 6). Furthermore, Lec2 CHO cells modified with α2-6-linked natural sialic acid did not exhibit significant effects on osteoclast formation presumably due to the relatively low affinity

of **α2-6-6** for Siglec-15 compared to **α2-6-10** (Supplementary Fig. 4).

## Discussion

The use of molecular diagnostic techniques for the detection of biological markers in the genome or proteome has dramatically increased over the last several decades, which has stemmed from the advancements in molecular biology techniques and the miniaturization of high-throughput microarrays[48,49]. Microarrays are widely used to study disease-specific gene mutations or protein expression and to correlate these expression signatures with disease progression. As a result, clinicians are increasingly able to treat patients according to their individual gene and/or protein expression profiles, known as personalized medicine. While microarrays comprised of nucleic acid or peptides/proteins are well-established and accessible for common users, glycan microarray technology remains highly specialized[50].

The in-solution, cell-based glycan array platforms described here are developed to address this unmet need. By combining STs, ST6Gal1 and ST3Gal4, and α1-3FT with Lec2 CHO cells possessing a narrow and relatively homogeneous repertoire of N-linked glycoforms, cell arrays displaying sialosides, fucosides and their structurally related analogs can be assembled easily using instruments available in most biology labs. With the necessary glycosyltransferases and nucleotide sugar donors prepared in advance, it only requires 2–3 h to fabricate a cell-based glycan array displaying >30 distinct glycan epitopes in a 96-well plate. By contrast, it takes days to weeks to assemble a conventional glycan array using comparable chemoenzymatic chemistry because each individual glycan to be printed needs to be synthesized and purified[4,7]. Importantly, the diversity of cell-based glycan array platforms can be significantly expanded by including the recently developed CHO and HEK cell lines with simplified surface glycans via precision genome editing as foundation platforms[51].

Using conventional glycan microarray technology, high-affinity and selective glycan ligands have been identified for six out of 15 human Siglecs, which involves the synthesis of hundreds of sialylated oligosaccharides bearing unnatural substituents[35–38]. The cell-based array technique described here has proven to be a simpler alternative for this endeavor. By using STs to install unnatural sialic acids directly onto the cell surface in a linkage-specific manner, the lengthy synthesis of sialylated oligosaccharides is avoided. Because sialylation reactions are performed on the cell surface, significantly fewer materials are required to produce the desired structures.

Recently, Boltje and co-workers reported an alternative method to discover Siglec ligands directly on the cell surface[52,53]. This method involved metabolic labeling using an alkyne functionalized ManNAc precursor followed by cell-surface CuAAC. Although a few high-affinity ligands were discovered by this approach, none were specific for a particular Siglec. This is likely due to the lack of linkage specificity of this

**Fig. 4** Cell-based glycan arrays for screening Siglec-15 ligands. **a** The workflow for the construction of cell-based glycan arrays on the cell surface of Lec2 cells using a chemoenzymatic strategy for screening specific and high-affinity Siglec-15 ligands. Cells were first treated with ST6Gal1 and the alkyne functionalized CMP-sialic acid analog **7**, followed by reacting with a library of azide-containing molecules via the biocompatible CuAAC to form a cell-based glycan array. The modified cells were incubated with Siglec-15-Fc pre-complexed with Anti-human IgG APC, then analyzed by flow cytometry. **b** Flow cytometry data presented in bar graph. Fold difference in MFI of Siglec-15 Fc binding was determined by comparison with the negative control (no transfer of **7**) ($n = 3$, error bars are reported as SEM). **c** Structures of CMP-SiaA13 (**10**) and CMP-SiaA31 (**11**) and resulting glycan structures after sialyltransferase-mediated cell-surface glycosylation. **d** Siglec-15-Fc binding after the one-step labeling of Lec2 CHO cells using ST6Gal1, ST3Gal1 or ST3Gal4-mediated transfer of **10** or **11**, respectively. Fold difference in MFI of Siglec-15 Fc binding was determined by comparison with the negative control ($n = 3$, error bars are reported as SEM). **e** Evaluating the specificity of α2-6-linked Siglec-15 ligands, **α2-6-10** and **α2-6-11** ($n = 3$, error bars are reported as SEM). **f** Evaluating the specificity of α2-3-linked Siglec-15 ligands, **α2-3-10** and **α2-3-11**($n = 3$, error bars are reported as SEM)

approach. Most Siglecs differentially bind to sialosides based on their Neu5Ac linkage. Metabolic labeling leads to incorporation of unnatural sialic acids in both α2-3- and α2-6-linkages, largely complicating the downstream Siglec binding studies. By contrast, the cell-based microarray described here utilized STs to install unnatural sialic acids directly on cell-surface glycans linkage specifically.

Siglec-15 is constitutively expressed in osteoclasts. During the RANKL-induced osteoclast differentiation, Siglec-15 is considerably upregulated[39–42]. Several antibodies targeting RANKL

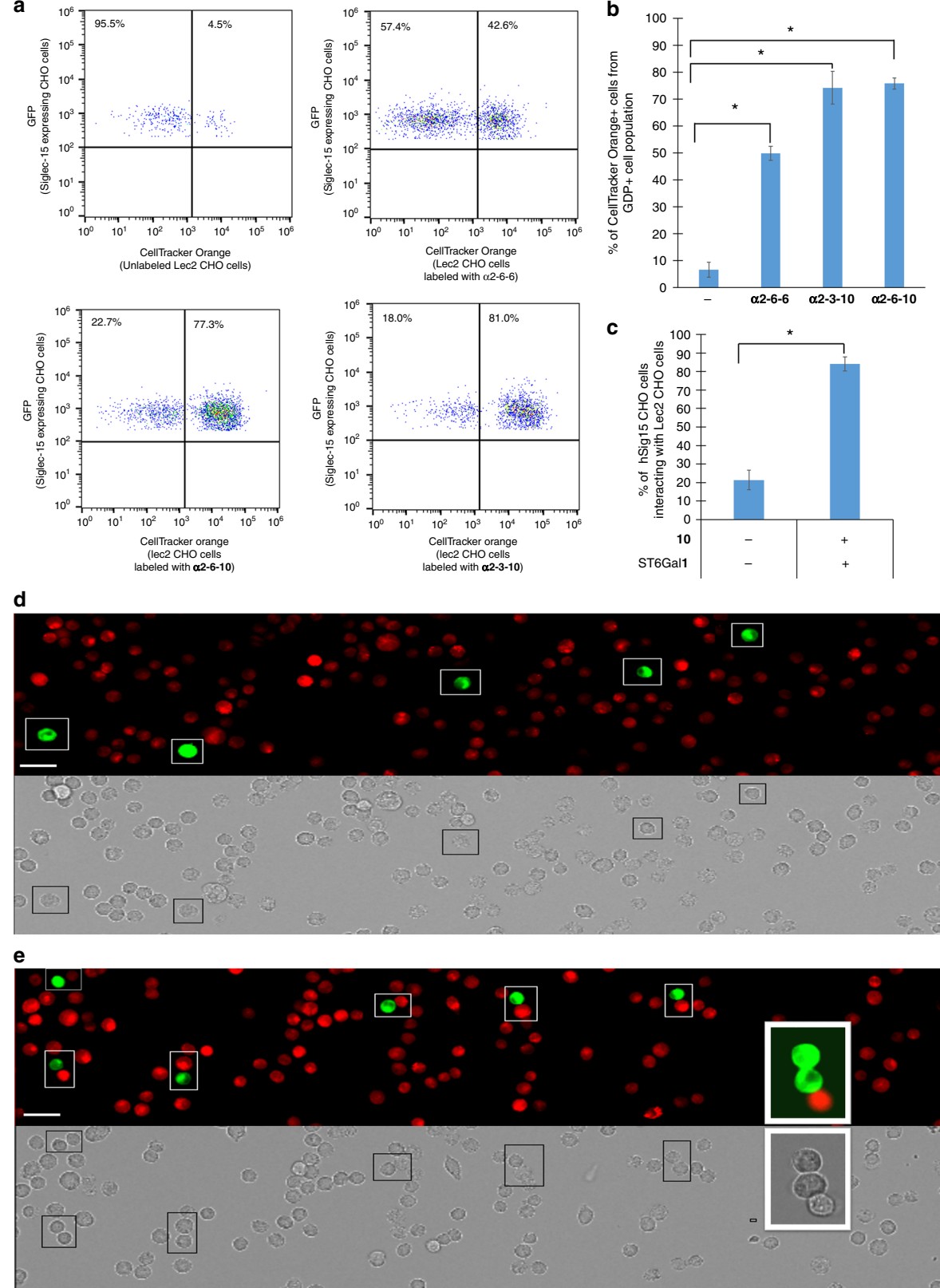

are FDA-approved, or are in late stage clinical trials for the treatment of osteoporosis and cancer-induced bone loss, and there are keen interests in targeting Siglec-15 for the same therapeutic applications[54–56]. In fact, Tremblay and co-workers discovered a monoclonal antibody for Siglec-15 that impaired osteoclast differentiation both in vitro and in vivo[46]. Exposure of human HOPs during differentiation to Lec2 CHO cells displaying the high-affinity Siglec-15 ligand (α2-6-10) impaired osteoclast formation, suggesting that this ligand may serve as the lead compound to be further optimized for clinical evaluation. While our method serves as a useful tool for the high-throughput screening of high-affinity glycan ligands for Siglecs and other GBPs, cells may display a range of glycans containing the modified monosaccharide. For the future evaluation of its therapeutic potential, isolation, purification, characterization, and validation of the high-affinity Siglec-15 ligand are required. Currently, we are in the process of using the cell-based glycan array technique to identify specific and high-affinity ligands for other Siglecs.

## Methods

**Cell culture conditions**. Lec2 and Lec8 CHO cells were grown in monolayer in alpha-Minimum Essential medium (α-MEM) (Invitrogen) supplemented with 10% fetal bovine serum (FBS) (Sigma-Aldrich). In all cases, cells were incubated in a 5.0% carbon dioxide, water-saturated incubator at 37 °C.

**Synthesis of 10 and 11**. CMP-SiaPoc (**7**) (10 mM) and azide A13 or A31 (12 mM) were dissolved in 1:1 DMSO/$H_2O$ containing premixed 3-[4-({bis[(1-*tert*-butyl-1H-1,2,3-triazol-4-yl)methyl]amino}methyl)-1H-1,2,3-triazol-1-yl]propanol (BTTP)-$CuSO_4$ complex (2 mM $CuSO_4$, 4 mM BTTP) and 5 mM freshly prepared sodium ascorbate and agitated for 3 h at 30 °C. Reaction completion was monitored by TLC and LCMS. Upon consumption of **7**, the reaction was quenched with EDTA (4 mM). The mixture was concentrated in vacuo and crude reaction products were purified by Bio-Gel P2 gel filtration chromatography eluted with $NH_4CO_3$ (50 mM). Only the fractions containing the product were collected and lyophilized.

**General procedure of cell-surface in situ glycosylation**. In situ fucosylation of cells was performed as previously described[57]. Briefly, cells were resuspended in 100 μL HBSS buffer containing 20 mM $MgSO_4$, 3 mM HEPES, 0.1% FBS, 50 μM GDP-fucose analogs (**1−5**), 30 mU α1-3FT. After incubation at 37 °C for 10 min, the cells were washed three times with PBS. In situ sialylation of cells was performed as previously described[18,23]. Briefly, cells were washed three times with PBS and resuspended in 100 μL serum-free α-MEM containing 0.65 μL BSA (2 mg/mL), 0.65 μL Shrimp Alkaline phosphatase (1000 U/mL, New England BioLabs), 31 μL of 1.5 M sucrose, 4.2 μL ST6Gal1[17] or ST3Gal4[16] (1 mg/mL) and 4 μL CMP-sialic acid analogs (**6−11**) (10 mM) with a density of 0.5−1.0×10⁶ /100 μL in 96-well plates at 37 °C for 1 h. The treated cells were then washed three times with PBS.

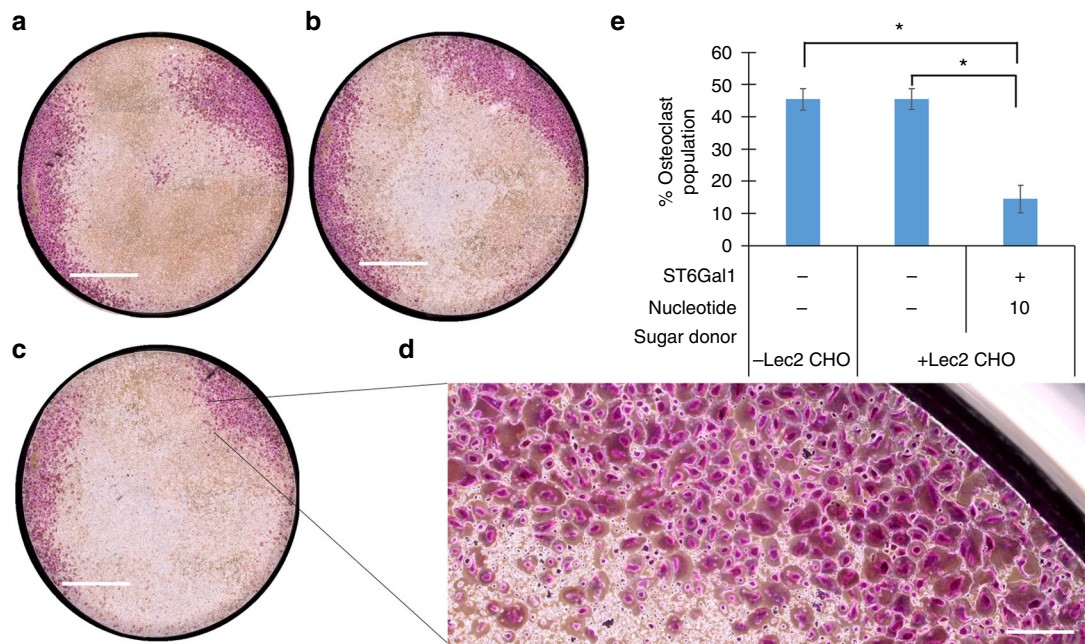

**Fig. 6** Siglec-15 ligands impair osteoclast differentiation. Osteoclast differentiation was induced by the addition of hM-CSF and RANKL to the plated HOPs. On day 5, Lec2 CHO cells labeled with the high-affinity Siglec-15 glycan ligand, α**2-6-10**, or the native ligand, α**2-6-6**, were added to the plated cells. Quantitative analysis of osteoclast formation by a TRAP-staining assay was performed on day 7 on cells with no addition of Lec2 CHO cells (scale bar: 4 mm) (**a**), with the addition of unlabeled Lec2 CHO cells (scale bar: 4 mm) (**b**), and with the addition of Lec2 CHO cells labeled with α**2-6-10** (scale bar: 4 mm) (**c**). **d** Zoomed-in regions of osteoclast population (scale bar: 200 μm). **e** Percentage of well containing osteoclast populations (area of purple/entire well area × 100%; $n = 3$, error bars are reported as SEM, significant difference compared with control was assessed using Student's $t$test (*$P < 0.01$))

**Fig. 5** High-affinity Siglec-15 ligands bridge cell interactions. Lec2 CHO cells displaying high-affinity Siglec-15 ligands or unmodified Lec 2 CHO cells were stained with CTO and mixed with Siglec-15-expressing CHO cells that express extracellular Siglec-15 fused to GFP in the intracellular domain. The formation of cell clusters was determined by double staining and analyzed by flow cytometry and fluorescence microscopy. **a** Dot plot displaying GFP+ population (Siglec-15-expressing CHO cells) that are interacting with CTO-stained Lec2 CHO cells. **b** Percentage of GFP+ population (Siglec-15-expressing CHO cells) that interact with CTO-stained Lec2 CHO cells ($n = 3$, error bars are reported as SEM, significant difference compared with control was assessed using Student's $t$-test (*$P < 0.0001$)). **c** Percentage of Siglec-15-expressing CHO cells (GFP+) that interact with CTO-stained Lec2 CHO cells by fluorescence microscopy ($n = 3$, error bars are reported as SEM, significant difference compared with control was assessed using Student's $t$-test (*$P < 0.0001$)). **d** Fluorescence microscopy images of Siglec-15-expressing CHO cells (GFP+) and CTO-stained, unlabeled, Lec2 CHO cells (scale bar: 40 μm). **e** Fluorescence microscopy images of Siglec-15-expressing CHO cells (GFP+) interacting with CTO-stained Lec2 CHO cells labeled with α**2-6-10** (scale bar: 40 μm)

**General procedure of cell-surface CuAAC**. Alkyne-bearing cells, prepared as described above using the CMP-sialic acid analog **7** (CMP-SiaPoc), were reacted with 1 μL azide (5 mM stock) in a 100 μL PBS containing premixed BTTPS-CuSO$_4$ complex (75 μM CuSO$_4$, 450 μM BTTPS) and 2.5 mM freshly prepared sodium ascorbate for 5 min. The Cu(I)-catalyzed azide-alkyne cycloaddition was quenched by adding bathocuproine sulfonate (BCS, 1 mM).

**Cell-surface lectins and antibody staining**. Cells subjected to in situ fucosylation were assessed for binding of Le$^X$ lectins as follows. For SSEA-1, the cells were incubated with mouse monoclonal anti-SSEA-1 (5 μg/mL, R&D systems, Inc.) in 100 μL PBS with 2% FBS for 30 min followed by staining with PE-conjugated anti-mouse-IgM (5 μg/mL, Jackson ImmunoResearch Laboratories, Inc.) in 100 μL PBS with 2% FBS for 40 min. For AAL and Lotus A, the cells were incubated with Lotus A-FITC (0.1 mg/mL, EY Laboratories, Inc.), or AAL-FITC (10 μg/mL, Vector Laboratories) in 100 μL PBS with 2% FBS for 40 min. Cells were then washed three times with PBS, and resuspended in PBS with 2% FBS and 2 mM EDTA for flow cytometric analysis. Cells subjected to in situ sialylation were assessed for binding of LacNAc and sLacNAc lectins as follows. For ECA and SNA-I, the cells were incubated with ECA-FITC (1 μg/mL, EY Laboratories), or SNA-I-FITC (1 μg/mL, EY Laboratories) in 100 μL PBS with 2% FBS on ice. For MAL-II, the cells were incubated with MAL-II-FITC (1 μg/mL, EY Laboratories) in 100 μL PBS with 2% FBS and 1 mM CaCl$_2$ for 30 min on ice. Cells were then washed three times with PBS, and resuspended in PBS with 2% FBS and 2 mM EDTA for flow cytometric analysis.

**Cell-surface binding of Siglec-Fc chimeras**. Siglec-Fc chimeras containing the N-terminal Ig domains fused to the Fc region of human IgG1 were prepared as described previously[35]. Labeled Lec2 CHO cells were resuspended in 100 μL of recombinant Siglec-Fc chimera supernatants pre-complexed with anti-human IgG APC (1:40 dilution, 200 μg/mL). Briefly, 2.5 μL of anti-Human IgG APC (200 μg/mL) was added to 50 μL of recombinant Siglec Fc chimera supernatant and incubated in the dark at room temperature for 15 min followed by dilution with 50 μL of PBS containing 1% BSA. Cells resuspended in this solution were incubated for 30 min on ice before being washed three times with PBS, and resuspended in PBS with 2% FBS and 2 mM EDTA for flow cytometric analysis.

**Cloning and expression of Siglec-15 in CHO cells**. Siglec-15 contains a lysine residue in its transmembrane region that pairs with Dap-12. Since Dap-12 is not expressed in standard cell lines (e.g. CHO), we took another approach to expressing the extracellular portion of human Siglec-15 as a transmembrane portion of Siglec-15 by cloning a chimeric protein consisting of amino acids 1−260 of human Siglec-15 with the transmembrane and cytoplasmic tail of human CD22 fused to a C-terminal eGFP. This chimeric protein was cloned into pcDNA5/FRT/TOP/V5/His (Invitrogen) using the *Nhe*I and *Age*I restriction enzymes. CHO Flp-in cells (Invitrogen) cells stably transfected with this vector according to the manufacturer's protocol. After selection of cells with Hygromycin-B for 2 weeks, all cells were found to be expressing high levels of GFP by flow cytometry.

**Lec2 CHO cell and Siglec-15-expressing CHO cell clustering**. For flow cytometry analysis, Lec2 CHO cells were labeled with CellTracker orange (CTO) following the manufacturer's protocols (ThermoFisher Scientific), sialylated (as described above) and mixed with hSig15-expressing CHO cells that also express GFP in a ratio of 100:1 for 20−30 min on ice prior to analysis by flow cytometry. The GFP+ cell population was analyzed for CTO staining and interacting cells were defined as double stained. For fluorescence microscopy analysis, Lec2 CHO cells stained with CTO were sialylated as described above, mixed with hSig15-expressing CHO cells that also express GFP in a ratio of 100:1 for 20−30 min on ice prior to plating and incubation at 37 °C for 10 min to allow adherence. Plates were gently rinsed once with PBS and cell clustering was imaged by fluorescence microscopy. Images of each sample were analyzed for cell−cell interactions. The number of hSig15 CHO cells that were interacting with Lec2 CHO cells was counted and expressed as a percentage of the total number of hSig15 CHO cells analyzed.

**Osteoclast differentiation**. HOPs (CD14+ peripheral blood mononuclear cells) were isolated from normal human peripheral blood mononuclear cells using EasySep Human Monocyte Isolation Kit (StemCell Technologies) following the manufacturer's instructions. HOPs were plated at $0.5×10^6$ cells/well in a 24-well plate with 1 mL media (αMEM supplemented with 1 mM sodium pyruvate and 10% FBS). To stimulate osteoclast formation, 25 ng/mL of human macrophage colony-stimulating factor and 30 ng/mL of human RANKL (R&D Systems) were added. Cells were allowed to differentiate for 7 days with half media replaced every 3 days. On Day 5, Lec2 CHO cells ($0.1×10^6$ cells) sialylated as described above using glycosyl donor **10** were added to wells. Unlabeled Lec2 CHO cells were used as a negative control. On Day 7, cells were fixed and permeablized followed by staining for TRAP using the TRAP kit following manufacturer protocols (Sigma Aldrich). Osteoclasts were defined as TRAP+ with 3 or more nuclei.

**Data availability**. All data are available from the authors upon reasonable request.

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

## Acknowledgements

Financial supports from the NIH (GM093282 and GM113046 to P.W. and GM103390 to K.W.M.) and NSFC-Shandong Joint Fund for Marine Science Research Centers (U1606403) are gratefully acknowledged. We would like to thank Prof. J.C. Paulson for providing Siglec-Fc chimeras and for stimulating discussions, and Prof. Pamela Stanley for providing CHO Lec mutants.

## Author contributions

P.W. conceived the concept of cell-based glycan array and supervised the project. M.S.M. co-supervised the project and provided resources. J.G.B., H.J., K.W.M., M.S.M. and P.W. contributed to experimental design. K.W.M. provided reagents. J.G.B. and H.J. performed the experiments. J.G.B., M.S.M. and P.W. analyzed the data. J.G.B. wrote the manuscript, which was edited and approved by all authors.

## Additional information

**Competing interests:** The authors declare no competing financial interests.

