## [Peer Review File · Nature Communications]

Reviewers' comments:

Reviewer #1 (Remarks to the Author):

The manuscript by Briard et al describes a conceptually novel and interesting method for generating libraries of structurally diverse glycan ligand structures on surfaces of Lec2 CHO cells, with suppressed expression of native sialylated and fucosylated glycans. The authors used a clever combination of enzymatic and chemical modifications to derive a selective ligand for Siglec 15 and show that the introduction of Lec2 CHO mutants with the Siglec 15 ligand can inhibit osteoclast differentiation in Siglec 15 expressing CHO cells in co-culture.

This is a well designed and executed study providing an alternative to traditional glycan arrays, which enable similar studies but are more technically involved, as they typically include a careful purification and characterization of individual glycan structures introduced into the microarrays. I should be noted that, while the described approach is faster and requires less specialized instrumentation, it does eliminate the careful glycan characterization step. Presumably, the ligand responsible for the Siglec 15 activity will have to be eventually isolated, characterized and validated (presumably the Lec2 CHO α 2-6-10 cells display a range of glycans containing the active modification). Nonetheless, sometimes there is a value in sacrificing control for the sake of efficiency and throughput and, although the biologically active Siglec15 ligand was not validated in this study, the phenotypic osteoclast differentiation assay is reasonably convincing.

One concern that arises from this assay and the authors should address is that the natural Siglec15 ligand, α 2-6-6, shows no effect on osteoclast differentiation. This is unexpected.

Overall, this is an interesting new addition to the glycobiology toolbox.

Reviewer #2 (Remarks to the Author):

This is a nicely designed good paper describing a novel approach to investigating cell surface glycan-glycan binding proteins (GBPs) unlike conventional types of glycan arrays. The reviewer finds interest and recognizes that the paper should be acceptable for the publication in this journal. However, he also finds some concerns in basic aspects in the introductory part and in the first part of Results (Fig. 1 and 2). The application of the method using sugar nucleotide analogues (Fig. 3) is also intriguing and the finding of a siglec 15 ligand is significant.

I. 28: Glycans do not cover only eukaryotic cells but also prokaryotic cells. Modify the sentence appropriately.

I. 39: The authors mention a part of Glycoscience Roadmap (ref. 6), but it's not clear which part they refer to. Please specify the sentence or at least section in the ref.

I. 81: It is true that Lec2 mutant lacks CMP-transporter and thus is a good model cell to provide glycan acceptor scaffold, but the cell possesses strong activity of α 2-3ST. So, it is likely to produce α -2-3 sialylated glycans even if α 2-6ST is transfected. The authors should tell this information and explain what happened actually under such conditions. There is no relevant explanation in the manuscript.

Fig. 1: As is seen in the figure, CHO cells express mostly multi-antennary N-glycans while relatively few biantennary ones. This fact is well correlated to the observation when using LCA and PSA, both of which show strong binding to core-fucosylated bi-antennary N-glycans. The reviewer has interest on this point.

l. 113: ug to be "micro" g

Fig. 2C: In literature, ECA cannot bind to α 2-3 sialylated Lac(NAc), but the result shows there is stronger binding to ST3Gal4-transfected Cells than ST6 counterpart. Please explain.

l. 133: 3-4 orders of effective digit are not possible with any good analytical method to determine K_d/K_a . They are obviously overstatement based on calculation

Reviewer #3 (Remarks to the Author):

The manuscript by Briard et. al. titled "Cell-Based Glycan Arrays for Probing Glycan-Glycan Binding Protein Interactions" described a method to generate a library of differentially sialylated and/or fucosylated CHO-cells and used these cells as a discovery platform for Siglec-15 study. The manuscript is well presented and the results are of considerable significance. However, there are also concerns about this manuscript:

1. The authors claims this "cell-based glycan microarrays", however, what is presented in the manuscript is a relatively small library of rather limited structural diversity. It is also difficult to image the expansion of this approach to other classes of glycans.
2. The use of flow cytometry to analyze protein-glycan interactions at cell surface is not new. While this is a useful approach, it is generally not considered as a "glycan array" format. Although the term "array" can be loosely used for a library of anything, it usually indicates a single assay of the whole library simultaneously. This is not the case for normal flow cytometry format. Giving the number of different cells used in this study, it is not well justified to use the term "glycan array".
3. While the cell-format protein-glycan interaction analysis provide certain advantage over more artificial solid surfaces, it also brings significant quality control problem. While the engineering of cell surface glycans using glycosyltransferases is well known, the only way to confirm the glycan structures is through binding assay, which is less quantitative and the epitope homogeneity is always of question.
4. The approach is targeting N-glycome, whether the O-glycome is affected is not well established.
5. The use of Click chemistry to increase the diversity of the library is useful. Can the authors provide some rationale for the selection of the library and possible mechanism for higher affinity of "A13, A31 & A37" (Line 203)?

Reviewer 1

“It should be noted that, while the described approach is faster and requires less specialized instrumentation, it does eliminate the careful glycan characterization step. Presumably, the ligand responsible for the Siglec 15 activity will have to be eventually isolated, characterized and validated (presumably the Lec2 CHO α 2-6-10 cells display a range of glycans containing the active modification). Nonetheless, sometimes there is a value in sacrificing control for the sake of efficiency and throughput and, although the biologically active Siglec15 ligand was not validated in this study, the phenotypic osteoclast differentiation assay is reasonably convincing.”

We agree with the reviewer that isolation, characterization, and validation will eventually be required to confirm the precise chemical nature of the underlying glycan structure that comprises the Siglec-15 ligand. However, we feel that this endeavor is beyond the scope of the current manuscript because our intention is to primarily describe the approach for discovering ligands. Nevertheless, as this is a valid point the reviewer brings up, we have included the following statement in the revised manuscript: “While our method serves as a powerful tool for the high-throughput screening of high-affinity glycan ligands for Siglecs and other GBPs, cells may display a range of glycans containing the modified monosaccharide. For the future evaluation of its therapeutic potential, isolation, purification, characterization, and validation of the high affinity Siglec-15 ligand is required.”

“One concern that arises from this assay and the authors should address is that the natural Siglec15 ligand, α 2-6-6, shows no effect on osteoclast differentiation. This is unexpected.”

We were also initially surprised by this result. One potential explanation could be that in this assay, the affinity of the native ligand is too low to generate a response, i.e. the inhibition of osteoclast differentiation. We have included in the revised manuscript, that: “Furthermore, Lec2 CHO cells modified with α 2-6-linked natural sialic acid did not exhibit significant effects on osteoclast formation presumably due to the relatively low affinity of **α 2-6-6** for Siglec-15 compared to **α 2-6-10**.”

Reviewer 2

l. 28: Glycans do not cover only eukaryotic cells but also prokaryotic cells. Modify the sentence appropriately.

We thank the reviewer for pointing this out. This sentence has been modified, to now read: “Glycans decorate the cell surface of both eukaryotes and prokaryotes and in mammalian cells

are involved in a variety of physiological processes in, including angiogenesis, fertilization, stem cell development, and neuronal development.”

l. 39: The authors mention a part of Glycoscience Roadmap (ref. 6), but it's not clear which part they refer to. Please specify the sentence or at least section in the ref.

We thank the reviewer for bringing this up. We have now included the section that this quotation appears: “As highlighted in Glycoscience Roadmap (section 5.1.1), these microarrays have found “wide utility for interrogating binding specificities of a diverse range of GBPs, determining dissociation constants and dissecting binding energies, and analyzing multivalent and hetero-ligand binding”.

l. 81: It is true that Lec2 mutant lacks CMP-transporter and thus is a good model cell to provide glycan acceptor scaffold, but the cell possesses strong activity of a2-3ST. So, it is likely to produce a-2-3 sialylated glycans even if a2-6ST is transfected. The authors should tell this information and explain what happened actually under such conditions. There is no relevant explanation in the manuscript.

We are confused by this comment because no transfections of sialyltransferases were carried out. Instead, the enzymes were applied to cells in order to exclusive sialylate the outside of the cells. In short, we used ectopic *addition* of enzyme, rather than *expression* of enzyme. We feel this point was clearly made in the manuscript.

l. 113: ug to be "micro" g

This has been changed.

Fig. 2C: In literature, ECA cannot bind to a2-3 sialylated Lac(NAc), but the result shows there is stronger binding to ST3Gal4-transfected Cells than ST6 counterpart. Please explain.

As described above, transfections of the STs were not carried out. Rather, the cells were modified by *in situ* addition of the sialyltransferases and CMP-Sia. Nevertheless, the point is still valid about the basis of ECA binding to the cells modified by ST3Gal4 better than that modified by ST6Gal1. First, is notable that ST3Gal4-treatment indeed reduces ECA binding compared to the untreated cells, indicating that ST3Gal4 is active. Second, ECA does not bind equally to all terminal LacNAc structures (<https://www.ncbi.nlm.nih.gov/pubmed/17805962>). There is a possibility that the preferred acceptor for ST3Gal4 on cells is bound poorly by ECA. Therefore, modification by ST3Gal4 will not reduce ECA binding significantly.

l. 133: 3-4 orders of effective digit are not possible with any good analytical method to determine Kd/Ka. They are obviously overstatement based on calculation

We thank the reviewer for pointing out this good point. We have revised the number of significant digits to 2.

Reviewer 3

The authors claims this “cell-based glycan microarrays”, however, what is presented in the manuscript is a relatively small library of rather limited structural diversity. It is also difficult to image the expansion of this approach to other classes of glycans.

Although we only reported fucosylated and sialylated glycan arrays created on the surface of Lec2 CHO cells in the manuscript, the diversity of cell-based glycan array platforms can be significantly expanded in the following ways:

- (1) As pointed out in the manuscript, “Importantly, the diversity of cell-based glycan array platforms can be significantly expanded by including the recently developed CHO and HEK cell lines with simplified surface glycans via precision genome editing as foundation platforms.^{12, 51}” These cells are known to possess monoantennary or biantennary N-glycans, after in situ glycosylation (e.g. sialylation) glycan arrays displaying modified glycans with different multivalency can be created.
- (2) Using biorthogonal chemistry, unnatural functional groups can be introduced to increase structural diversity. This point has also been demonstrated in the manuscript.
- (3) Using additional recombinant glycosyltransferases. For example, ST3Gal1 can be used to install natural or unnatural sialic acid onto O-glycans. By combining ST3Gal1 and Lec 2 CHO cells, we can create a cell-based array displaying Sia α 2,3Gal β 1,3GalNAc α Ser/Thr and its structurally related derivatives.

The use of flow cytometry to analyze protein-glycan interactions at cell surface is not new. While this is a useful approach, it is generally not considered as a “glycan array” format. Although the term “array” can be loosely used for a library of anything, it usually indicates a single assay of the whole library simultaneously. This is not the case for normal flow cytometry format. Giving the number of different cells used in this study, it is not well justified to use the term “glycan array”.

We agree that the “array” presented in this article is not a conventional glycan array in which glycans are generally appended to a solid support and GBP-binding is measured by fluorescence microscopy. In our method, each glycan is presented on a cell-surface rather than a slide. We understand the concern about whether this method encompasses the traditional definition of an “array”. There is a distinct glycan structure in each well of a 96-well plate generating an “array” of glycan structures that can then be analyzed for GBP-interaction by flow cytometry. Therefore, we argue that this format is an array, but given the obvious differences to a traditional array and taken into the consideration of the reviewer’s comments, we have now redefined in the revised manuscript this new approach as a “in-solution, cell-based glycan array”.

While the cell-format protein-glycan interaction analysis provide certain advantage over more artificial solid surfaces, it also brings significant quality control problem. While the engineering of cell surface glycans using glycosyltransferases is well known, the only

way to confirm the glycan structures is through binding assay, which is less quantitative and the epitope homogeneity is always of question.

We agree that that quality control of our solution-based glycan array cannot be done in the same manner as when working with chemically-defined libraries of glycans. Cell-surface glycosylation is inherently heterogeneous, which is both an advantage and disadvantage of this approach. It is an advantage because a larger set of underlying glycan structures can be explored. Conversely, this heterogeneity is a disadvantage because it will make finding the precise species serving as ligand for the glycan-binding protein more challenging. However, for the approach described in this manuscript, using Lec2 CHO cells, we argue that there is no reason to suspect that the glycosylation of Lec2 CHO cells will be reasonably variable from day-to-day and from laboratory to laboratory. Furthermore, by carrying out kinetics to ensure that saturating amounts of enzyme and substrate are used, as was done and highlighted in our studies, this maximizes the probability that the density of modified glycans presented on the cell surface is consistent from experiment-to-experiment. Therefore, based on these two points we argue that issues surrounding quality control are not a major concern with our approach. Indeed, in numerous independent trials, the binding of Siglec-15 to cells modified with $\alpha 2-6-10$ was reproducible in magnitude compared to the parent sialic acid modification ($\alpha 2-6-6$). Finally, we wish to point out that fluorescence, as measured by flow cytometry, is at least as quantitative - if not more quantitative - than traditional solid support-based glycan arrays that use scanners to detect fluorescence.

The approach is targeting N-glycome, whether the O-glycome is affected is not well established.

As demonstrated by Boons and coworkers previously, recombinant ST6Gal1 only modifies N-linked glycans, leaving O-glycans untouched (*Angew. Chem. Int. Ed.* **2013**, 52, 13012 –13015). ST3Gal4 is known to be involved in the biosynthesis of sialyl-Lewis X on O-linked glycans. But the O-glycans in Lec2 CHO cells are primarily un-elongated core-1 structures and, therefore, cannot be modified by recombinant ST3Gal4.

To modify O-linked glycans on the cell surface to create cell arrays, ST3Gal1 can be used. As shown in **Supplementary Fig. 4**, ST3Gal1 is able to transfer a biotinylated sialic acid analog onto the surface of Lec 2 CHO cells.

The use of Click chemistry to increase the diversity of the library is useful. Can the authors provide some rationale for the selection of the library and possible mechanism for higher affinity of “A13, A31 & A37” (Line 203)?

There is no reported crystal structure of Siglec 15, therefore, the rational design of a high affinity ligand is impossible. We have included in the revised manuscript: “The library was chosen to contain structures with a variety of different functional groups to cover a range of chemical space.” Nevertheless, adding hydrophobic groups at the 5- and 9-position of NeuNAc has been used by several groups to increase affinity to Siglecs in the absence of structural information. Without a crystal structure of Siglec-15, it is very difficult to provide a precise mechanism for the high affinity associated with A13, A31 & A37, except that we can speculate that it is extremely likely that these hydrophobic substituents are exploiting hydrophobic grooves on the surface of Siglec-15.

Reviewers' Comments:

Reviewer #1:

Remarks to the Author:

The revised manuscript addressed my concerns. I have no further comments

Reviewer #2:

Remarks to the Author:

The reviewer confirmed that all the points raised were appropriately corrected, and the revised manuscript looks better so that he is convinced that it should be now fully acceptable in the journal with no further modification.

Reviewer #3:

Remarks to the Author:

The authors have reasonably addressed the concerns by reviewers and publication is recommended.

There are no issues from the referees to be addressed.